# Does information structuring improve recall of discharge information? A cluster randomized clinical trial

**Victoria Siegrist**[1,2], **Rui Mata**[2], **Wolf Langewitz**[3], **Heike Gerger**[4,5], **Stephan Furger**[6], **Ralph Hertwig**[7], **Roland Bingisser**[1] *

**1** Emergency Department, University Hospital Basel, Basel, Switzerland, **2** Center for Cognitive and Decision Sciences, University of Basel, Basel, Switzerland, **3** Department of Psychosomatic Medicine–Communication in Medicine, University Hospital Basel, Basel, Switzerland, **4** Clinical Psychology and Psychotherapy, Faculty of Psychology, University of Basel, Basel, Switzerland, **5** Department of General Practice, Erasmus MC University Medical Center, Rotterdam, The Netherlands, **6** Translational Research Center, University Hospital of Psychiatry, University of Bern, Bern, Switzerland, **7** Center for Adaptive Rationality, Max Planck Institute for Human Development, Berlin, Germany

* roland.bingisser@usb.ch

**Data Availability Statement:** Anonymized data are available on OSF under the following link: https://osf.io/84q3r/.

## Abstract

### Objectives

The impact of the quality of discharge communication between physicians and their patients is critical on patients' health outcomes. Nevertheless, low recall of information given to patients at discharge from emergency departments (EDs) is a well-documented problem. Therefore, we investigated the outcomes and related benefits of two different communication strategies: Physicians were instructed to either use empathy (E) or information structuring (S) skills hypothesizing superior recall by patients in the S group.

### Methods

For the direct comparison of two communication strategies at discharge, physicians were cluster-randomized to an E or a S skills training. Feasibility was measured by training completion rates. Outcomes were measured in patients immediately after discharge, after 7, and 30 days. Primary outcome was patients' immediate recall of discharge information. Secondary outcomes were feasibility of training implementation, patients' adherence to recommendations and satisfaction, as well as the patient-physician relationship.

### Results

Of 117 eligible physicians, 80 (68.4%) completed the training. Out of 256 patients randomized to one of the two training groups (E: 146 and S: 119) 196 completed the post-discharge assessment. Patients' immediate recall of discharge information was superior in patients in the S-group vs. E-group. Patients in the S-group adhered to more recommendations within 30 days ($p$ = .002), and were more likely to recommend the physician to family and friends (p = .021). No differences were found on other assessed outcome domains.

**Funding:** This project was funded by the Swiss National Science Foundation (CR31I3_159841). The funders had no role in study design, data collection and analysis, decision to publish, or preparation of the manuscript.

**Competing interests:** The authors have declared that no competing interests exist.

## Conclusions and practice Implications

Immediate recall and subsequent adherence to recommendations were higher in the S group. Feasibility was shown by a 69.6% completion rate of trainings. Thus, trainings of discharge information structuring are feasible and improve patients' recall, and may therefore improve quality of care in the ED.

## 1 Introduction

The need to communicate effectively is pervasive in healthcare [1], and it is of utmost importance in the acute care setting such as the emergency department (ED) [1]. Patients are typically discharged within few hours once life-threatening conditions have been excluded, but serious conditions may still be under investigation in the outpatient setting. An effective communication has an impact on both physicians and patients: Physicians' communicative behaviour was found to be linked to the charge of malpractice claims [2], and patients' health outcomes were shown to be related to the quality of the communication with physicians [3]. Furthermore, patients need to be well-informed to be able to follow treatment recommendations [4]. Previous studies have found that patients only recall a fraction of the information given [4–7]. Therefore, the importance of information delivery at ED discharge, and its downstream implications for recall and patient outcomes have been emphasized [8, 9].

Several communication strategies have been found to improve recall of medical information, for instance, provision of written information [10–12]. However, the provision of written information is difficult in a busy ED environment and has not uniformly shown benefits [13–15], possibly due to the fact that written discharge information often exceed patients' health literacy and their levels of reading and understanding skills [16, 17].

Among the verbal communication strategies, information structuring seems to be the most promising strategy for improving information recall: Several studies have found improved recall in proxy-patients if medical information was explicitly structured [18–20]. These studies used a specific way of information structuring, the "book metaphor" technique: Discharge information was provided with an initial "table of contents" followed by "chapter headings" [21]. The use of the mnemonic "InFARcT" (**In**formation on diagnosis; **F**ollow-up; **A**dvice on self-care; **R**ed flags; **c**omplete **T**reatment) was shown to significantly improve recall [18–20]. Structuring the content of information given can take many forms; in pre-medication visits [22] the topics to be dealt with differ from discharge communication from patient to patient. So far however, improved recall of discharge information by explicitly structuring discharge information has only been shown in proxy patients (i.e. student populations), and never in ED patients.

We therefore designed a cluster randomized clinical trial to investigate the effects of information structuring on patients' recall, adherence to recommendations, and satisfaction. As a control group, some clusters of physicians were trained using empathy skills, ensuring a credible control group that does not focus on conveying information per se. Empathy skills seem particularly relevant in acute care because ED patients suffer from high levels of stress and anxiety [23, 24]. Furthermore, several studies found positive effects on patients' outcomes if physicians used empathy skills [25–27]. Consequently, we compared the effects of training physicians' information structuring (S) skills with training their empathy (E) skills. We hypothesized that explicit information structuring using the book metaphor and the InFARcT mnemonic would be feasible, and would improve patients' information recall, their adherence to instructions [4], and patients' satisfaction [26, 27].

## 2 Methods

### 2.1 Design, setting and participants

This two-arm, cluster randomized controlled trial was conducted at the ED of the University Hospital Basel, Switzerland. The study was approved by the local ethics committee "Ethikkommission Nordwest- und Zentralschweiz" (EKNZ 2014–379) on December 3, 2014 and the protocol was published on ClinicalTrials.gov (NCT02468869). Physicians and patients were enrolled between April 1, 2015 and May 31, 2017. The registration for the clinical trial on ClinicalTrials.gov took place on October 12, 2015 because of changes in the study team. Nevertheless, this did not affect the study conduct or results. All authors confirm that all ongoing and related trials for this intervention are registered.

**2.1.1 Physicians.** As study physicians we included new residents starting at the ED of the University Hospital Basel. Physicians were clustered according to their first day at work (January 1st, April 1st, July 1st, and October 1st). Eight clusters of physicians were included (see Fig 1). Physicians were blinded regarding cluster randomization and the content of the other communication skills training. They gave written informed consent before undergoing three teaching modules of communication training (see section 2.2).

**2.1.2 Patients.** Patients were eligible if they presented to the ED with chest or abdominal pain and were discharged by a physician under study. Patients were not eligible if they met one of the following exclusion criteria: they did not provide written consent, were younger than 18 years old, non-German speaking, or had a diagnosis of dementia.

**2.1.3 Randomization.** For cluster randomization, we determined the content of the communication training for the first cluster of physicians using an electronic randomizer tool (randomizer.org) generating a random sequence, based on which each cluster of physicians starting shift-work at the ED was randomly assigned to one of the two communication trainings (E or S).

### 2.2 Interventions

Physicians received a communication training consisting of three distinct modules: i) *communication with ED patients* (group instruction of 30 minutes, identical for all clusters), ii) according to randomization: either *empathy training* (*ET* group instruction of 75 minutes) or *structure skills training* (*ST*; group instruction of 75 minutes), and iii) *feedback on the job* by an expert in communication (individual instruction of 15 minutes; detailed information in S1 File). The communication expert rated the physician's ability using the techniques conveyed earlier on a numeric rating scale from 1 to 6. Only after completing all three modules, physicians were able to participate in the study by performing audio-taped discharge communications.

### 2.3 Procedure

Patients presenting to the ED were screened for the main complaint of chest or abdominal pain using the web-based electronic health record. The electronic health record was fed with information from the attending physician, showing the collected information, such as main complaint, almost in real time. If patients were eligible, trained study personnel explained the study procedure and informed consent was obtained right after patient history was obtained. Patients were blinded to the communication training which their physician had received. Information on demographics, mental and physical health (12-Item Short Form Health Survey; SF-12) [28], anxiety and depression (Hospital Anxiety and Depression Scale; HADS-D)

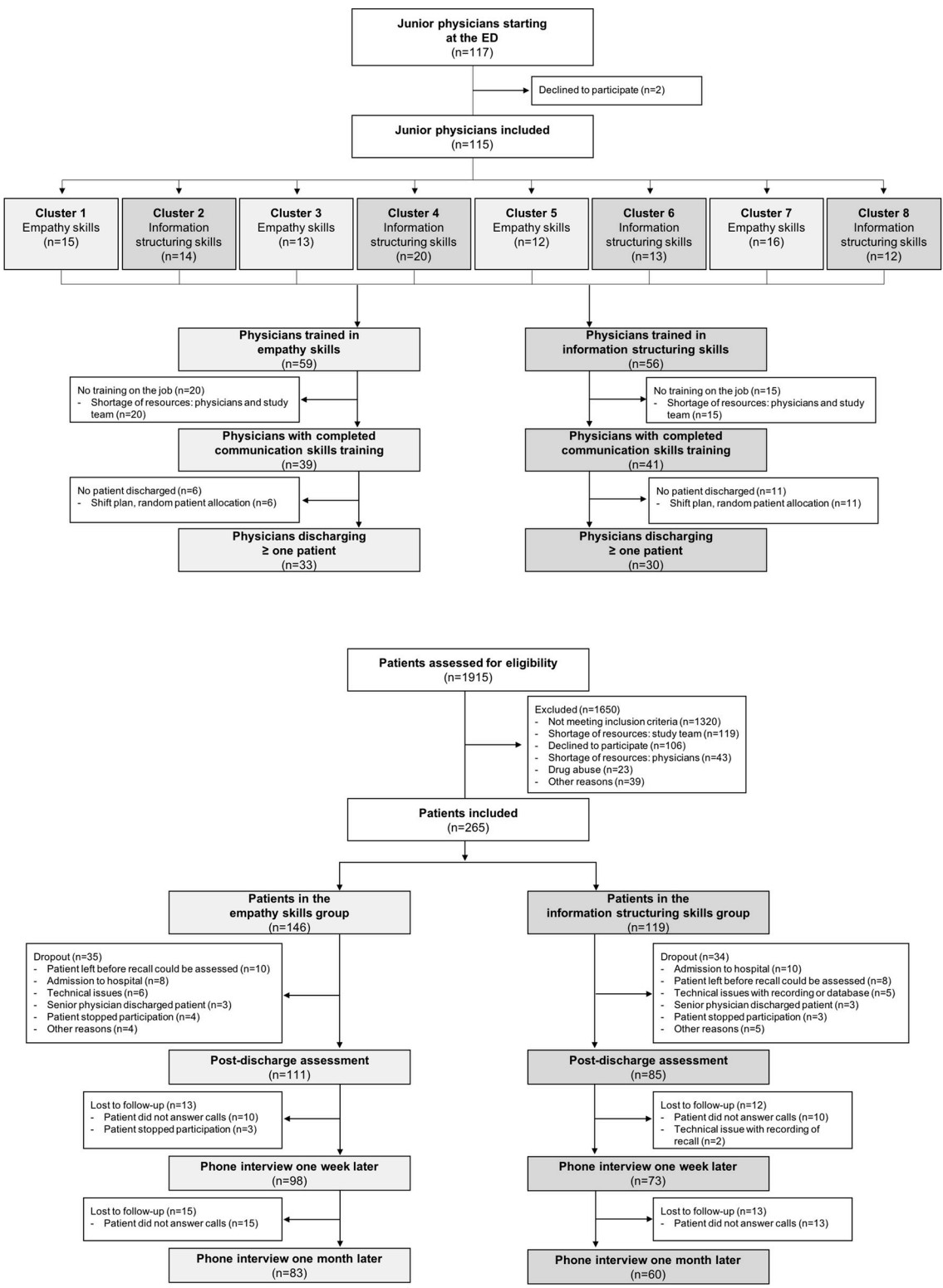

**Fig 1. Flow chart of trained physicians and included patients.**

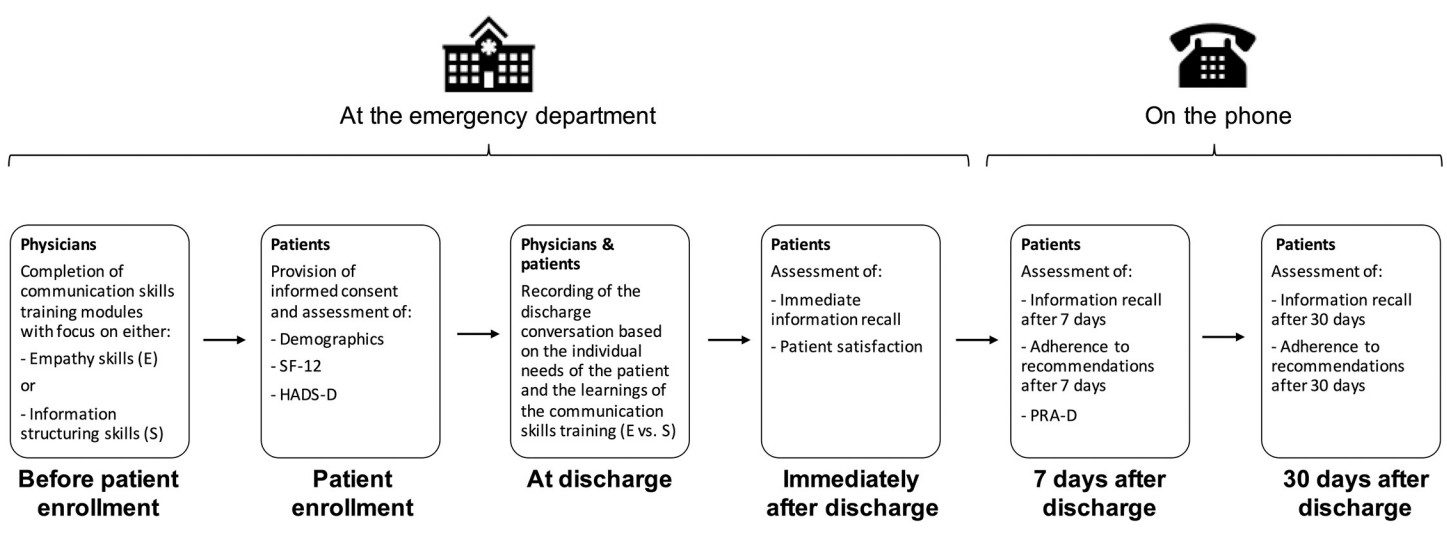

**Fig 2. Study overview.**

[29] was obtained right after informed consent was given. Data were recorded using the web-based software secuTrial® by study personnel.

After study personnel completed all surveys with the patient, the physician was given an audio recorder for recording the discharge communication. Immediately after the physician formally discharged the patient, trained study personnel assessed patient satisfaction and their recall of the discharge information. Before discharge, appointments were made for the follow-up calls 7 and 30 days later (see Fig 2).

### 2.4 Assessments

**2.4.1 Primary outcome: Immediate discharge information recall.** The primary outcome of the study was patients' *immediate recall* of discharge information as a function of physicians' communication training. This outcome was assessed right after the discharge communication was completed but before the patient left the hospital. The conversation between the researcher and the patient was recorded with an audio recorder. Recall performance of discharge information was assessed by the ratio between the number of utterances conveyed to the patient and the number of utterances recalled at discharge. Recall was assessed by asking patients the question: "Please share with me all the information that you recall from your discharge communication." After the last utterance was shared proactively by the patient, the interviewer would probe and ask "is there any additional information that you remember from your discharge communication?". The interview was stopped once the patient stated that there is nothing else that he or she recalls.

**2.4.2 Secondary outcomes: Feasibility, long-term recall (day 7 and 30), adherence, satisfaction, and patient-physician-relationship.** The *feasibility* of the training within the daily routines of ED was assessed by the percentage of physicians completing all three training modules during the study period. *Long-term recall* of discharge information was assessed on day 7 and 30 after discharge via telephone interviews. The researcher was calling the patient on the phone at the time they agreed upon before the patient left the ED at day 0. The recording of the conversation started after obtaining the consent of the patient regarding the recording. *Adherence*, also assessed via telephone interviews, was defined as the overall self-reported adherence in the first month relative to the total number of recommendations given by the

physician at discharge. Patients were asked "which recommendations did you follow since your visit?". *Patients' satisfaction* was measured immediately after discharge via telephone interview on four visual analogue scales. Each of these scales ranged from 0 to10 (0 indicating "not at all" and 10 indicating "very much") with the following dimensions: (1) "How easy was it to understand what your doctor said?"; (2) "How structured was the communication?"; (3) "How informative was the communication?"; and (4) "How strongly would you recommend the doctor to family and friends?". Additionally, 7 days after discharge, patients completed the German version of the Patient Reaction Assessment (PRA-D) [30, 31], which measures the *perceived quality of relationship between patient and physician*. PRA-D reflects patients' perceived quality of the informative (i.e. patient information index) and affective (i.e. patient affective index) behaviours of the physician, and patients' perceived ability to initiate communication (i.e. patient communication index).

**2.4.3 Characteristics and content of discharge information.**   Audio recordings of the discharge communication, patients' information recall, and patients' reports on adherence were transcribed with the software f4transkript. A detailed coding scheme was developed to specify all types of utterances that could be found in the transcripts (for detailed information see S1 Table). An utterance was defined as the smallest speech segment that expresses or implies a complete thought and that a coder can classify.

Based on the previous studies on the use of the book metaphor and the InFARcT mnemonic [14–16], we predefined seven main categories of utterances, which we called chapters according to the book metaphor: (1) explicit structure (e.g. introduction by providing title and chapter-headings of the information to follow), (2) **In**formation on diagnosis, (3) **F**ollow-up, (4) **A**dvice on self-care, (5) **R**ed-flags, (6) **c**omplete **T**reatment, and (7) other. After study completion, two independent coders, blinded to physicians' ID, cluster, and group, rated the transcripts of the discharge communication. Over 100 distinct utterances were identified and subsumed to the seven utterance categories.

In addition to the content of the discharge communication with respect to the 7 utterance categories, we compared the *duration of the discharge communication*, the *number of recommendations* given by the physician, and the *number of patient contributions* during discharge communication (i.e. questions asked and inputs given by the patient).

A manipulation check was conducted by rating each discharge communication based on the audio recording and the transcript by two blinded and independent raters regarding the amount of *empathic communication elements* and explicit *structure elements*. The two items indicating the presence of structure elements were "physician gives an outline" and "physician leads explicitly from one segment to another". The two items indicating empathy elements were "physician reacts when the patient shows an emotion" and "physician checks with the patient whether his concerns are clear". These were evaluated by rating each manipulation check with 0 (not given), 1 (partially given), 2 (completely given), or as "not applicable".

## 2.5 Sample size and statistical analyses

The study was powered at 80% (two-sided test, α-level of .05) to detect a difference in recall performance for two patient groups discharged either by a physician trained in empathy or structured discharge communication. With an estimated effect size of .4 (as deduced from a previously conducted laboratory experiment) a total sample size of 200 was required. We used an alpha level of .05 for all statistical tests.

Analyses were conducted using R (version 3.6.1.) in RStudio using the packages tidyverse [32] and lme4 [33] for calculating linear mixed effects models. For group comparisons, linear ANOVA analysis was used for continuous variables and Pearson's two-tailed chi-square test of

independence for categorical variables. The primary dependent variable was patients' immediate recall performance of discharge information. To investigate the recall performance of participants, linear mixed effects models were performed by taking the recall performance (i.e. number of utterances recalled by the patient) as the dependent variable and the group (i.e. E and S), as well as the utterances given by the physician during discharge as independent variables. This was done to investigate the effect of physician's communications trainings on patients' recall while controlling for the number of utterances given by the physician. Further, we included cluster ID (8 levels) and physician ID (63 levels) as nested random factor into the model to account for possible cluster-physician effects due to the randomization. This analysis, thus, reflects the total success of the training including the possibility that patients' recall may depend on the number of utterances given by the physician, which can be seen as a result of the respective training in itself. In addition, the analyses were repeated with patients' relative recall performance of discharge information at day 7 and 30. Effect sizes were labelled following Funder and Ozer's (2019) recommendations [34].

*Patients' adherence* to recommendations was the overall self-reported adherence to recommendations within one month once reported in absolute numbers and once in relative numbers (i.e. adherence divided by the number of recommendations given by the physician).

*Patient satisfaction* ratings showed ceiling effects and were therefore converted from interval into integer numbers. This permitted the computation of an ordinal model with the R package ordinal [35]. Each of the four satisfaction ratings were analysed with a cumulative link mixed model [35]. The models controlled for the random effects cluster and physician. Due to the skewed distribution of the patient satisfaction data we report medians and interquartile ranges (IQR) instead of means and standard deviations (SD).

# 3 Results

## 3.1 Participants

**3.1.1 Physicians.** A total of 117 physicians started their residency and were eligible during the study period, 115 signed consent (Fig 1). Four clusters of physicians were randomized to E, and four clusters to S. Cluster size ranged from 12 to 20 physicians. 80 physicians completed all three teaching modules during the study period, and 63 of them treated at least one patient within the context of this study (range: 1 to 19 patients per physician of which the discharge communication was recorded and included in the study). Physicians were on average 30.7 years old ($SD = 3.9$) and had previous experience of 2.9 years ($SD = 1.9$). Almost half of the physicians were women (47.6%). Physicians in groups E and S scored equally well in understanding and applying the communication skills taught in modules ii) empathy or structure skills, and iii) feedback on the job (Table 1). As physicians were cluster randomized to one of two communication trainings, there were no significant differences between physicians in E and S clusters in terms of age, sex, nationality, civil status, work experience, time spent on training, or proficiency in German. A detailed overview of physician's demographics can be found in Table 1.

**3.1.2 Patients.** In total, 1,915 patients were screened for eligibility (Fig 2). Of those, 1650 (80.0%) were not included because they did not meet inclusion criteria. A total of 265 patients were included in the study: 146 and 119 were treated by physicians of the E and S group, respectively. 196 (74.0%) patients completed the post-discharge assessment (111 in the E and 85 in the S group). Dropout rate from inclusion to post-discharge assessment was 26.0% (24.0% in the E and 28.6% in the S group). Patients had a mean age of 44.8 years ($SD = 16.4$), 42.9% were women (Table 1), and 76.5% reported to be German native speakers. With the exception of the physical health subscale of SF-12, there were no differences between the two

**Table 1. Physician and patient characteristics.**

| Physician Characteristics | Empathy | Structure | Overall | p-value[1] |
|---|---|---|---|---|
| | (N = 33) | (N = 30) | N = 63 | |
| Age, in years, mean (SD) | 30.4 (3.15) | 31.0 (4.56) | 30.7 (3.87) | .54 |
| Sex, N (%) | | | | .99 |
| *Male* | 17 (51.5%) | 16 (53.3%) | 33 (52.4%) | |
| *Female* | 16 (48.5%) | 14 (46.7%) | 30 (47.6%) | |
| Nationality, N (%) | | | | .12 |
| *Switzerland* | 7 (21.2%) | 13 (43.3%) | 20 (31.7%) | |
| *Germany* | 18 (54.5%) | 11 (36.7%) | 29 (46.0%) | |
| *Austria* | 2 (6.1%) | 4 (13.3%) | 6 (9.5%) | |
| *Other* | 6 (18.2%) | 2 (6.7%) | 8 (12.7%) | |
| Native language, N (%) | | | | .62 |
| *German* | 26 (78.8%) | 26 (86.7%) | 52 (82.5%) | |
| *Other* | 7 (21.2%) | 4 (13.3%) | 11 (17.5%) | |
| Civil status, N (%) | | | | .19 |
| *Single* | 25 (75.8%) | 26 (86.7%) | 51 (81.0%) | |
| *Divorced* | 0 (0.0%) | 1 (3.3%) | 1 (1.6%) | |
| *Married* | 8 (24.2%) | 3 (10.0%) | 11 (17.5%) | |
| Work experience, in years, mean (SD) | 2.8 (1.86) | 3.1 (1.92) | 2.9 (1.88) | .58 |
| *Unknown* | 1 | 0 | 1 | |
| Communication skills, mean (SD) | | | | |
| *Theory: Empathy vs structure skills* | 6.9 (1.52) | 7.1 (2.04) | 7.0 (1.77) | .73 |
| *Training on the job* | 4.7 (0.63) | 4.6 (0.83) | 4.6 (0.73) | .79 |
| **Patient Characteristics** | **Empathy** | **Structure** | **Overall** | **p-value** |
| | (N = 111) | (N = 85) | (N = 196) | |
| Complaint, N (%) | | | | **.019** |
| *Abdominal pain* | 63 (56.8%) | 33 (38.8%) | 96 (49.0%) | |
| *Chest pain* | 48 (43.2%) | 52 (61.2%) | 100 (51.0%) | |
| Sex, N (%) | | | | .76 |
| *Female* | 46 (41.4%) | 38 (44.7%) | 84 (42.9%) | |
| *Male* | 65 (58.6%) | 47 (55.3%) | 112 (57.1%) | |
| Age, in years, mean (SD) | 44.5 (17.37) | 45.1 (15.12) | 44.8 (16.39) | .80 |
| Education, N(%) | | | | .99 |
| *≤ Apprenticeship* | 59 (53.2%) | 46 (54.1%) | 105 (53.6%) | |
| *≥ High School* | 52 (46.8%) | 39 (45.9%) | 91 (46.4%) | |
| Nationality, N (%) | | | | .40 |
| *Swiss* | 77 (69.4%) | 55 (64.7%) | 132 (67.3%) | |
| *German* | 9 (8.1%) | 12 (14.1%) | 21 (10.7%) | |
| *Other* | 25 (22.5%) | 18 (21.2%) | 43 (21.9%) | |
| Native Language, N (%) | | | | .60 |
| *German* | 87 (78.4%) | 63 (74.1%) | 150 (76.5%) | |
| *Other* | 24 (21.6%) | 22 (25.9%) | 46 (23.5%) | |
| Civil Status, N (%) | | | | .27 |
| *Married* | 47 (42.3%) | 44 (51.8%) | 91 (46.4%) | |
| *Never married* | 43 (38.7%) | 29 (34.1%) | 72 (36.7%) | |
| *Divorced* | 17 (15.3%) | 8 (9.4%) | 25 (12.8%) | |
| *Seperated* | 2 (1.8%) | 4 (4.7%) | 6 (3.1%) | |
| *Widowed* | 2 (1.8%) | 0 (0.0%) | 2 (1.0%) | |

*(Continued)*

**Table 1.** (Continued)

| | | | | |
|---|---|---|---|---|
| SF-12, mean (SD) | | | | |
| *Physical Health* | 45.6 (10.20) | 48.7 (9.38) | 47.0 (9.94) | **.031** |
| *Mental Health* | 50.1 (10.30) | 51.1 (10.41) | 50.6 (10.33) | .50 |
| HADS, mean (SD) | | | | |
| *Anxiety* | 6.0 (3.69) | 5.9 (3.88) | 6.0 (3.77) | .91 |
| *Depression* | 4.0 (3.70) | 3.4 (3.23) | 3.7 (3.50) | .26 |

*Note.* Characteristics of the physicians who treated patients according to the training which they had received, and of the patients who completed post-discharge assessments.

[1] Continuous variables: Linear ANOVA; Categorical variables: Pearson's two-tailed chi-square test of independence.

groups in sex, age, nationality, proficiency in German, civil status, education, health related quality of life (SF-12), and anxiety and depression scores (HADS-D), indicating that the patient groups were comparable.

### 3.2 Discharge communication

**3.2.1 Characteristics.** Physicians' empathy and structuring skills during discharge communication were rated for each recorded discharge conversation. The manipulation check revealed that discharge communication was significantly more structured in S-trained physicians (mean = 2.6, *SD* = 1.3) compared to E-trained physicians (mean = 0.1, *SD = 0.3)* (difference = -2.54, 95% CI [-2.82, -2.25], $t(91.31) = -17.59$, $p < .001$). There were no significant differences in empathy ratings between E-trained (mean = 1.81, *SD* = 0.73) and S-trained (mean = 1.59, SD = 0.88) physicians (difference = 0.22, 95% *CI* [-0.01, 0.46], $t(162.36) = 1.89$, $p = .061$). Further, analyzing the recordings of the discharge conversations there was no difference between the two groups on the dimension's *duration of discharge communication* ($p = .345$), *number of patient contributions* (mean in group E = 9.77, mean in group S = 8.53, difference = 1.25, 95% *CI* [-1.34, 3.83], t(173.01) = 0.95, $p = .343$), and *number of recommendations* given by the physician (mean in group E = 4.44, mean in group S = 4.71, difference = -0.26, 95% *CI* [-0.82, 0.29], $t(193.51) = -0.95$, $p = .345$).

**Table 2. Discharge communication characteristics and content.**

| | Empathy | Structure | Overall | p-value[1] |
|---|---|---|---|---|
| | (N = 111) | (N = 85) | (N = 196) | |
| **Utterances by the physician, mean (*SD*)** | | | | |
| Explicit structure (book metaphor) | 0.1 (0.37) | 3.4 (1.98) | 1.5 (2.12) | < **.001** |
| Information on diagnosis | 14.1 (6.83) | 11.8 (5.85) | 13.1 (6.51) | **.012** |
| Follow-up | 6.7 (6.08) | 6.7 (4.86) | 6.7 (5.57) | .955 |
| Advice on self-care | 2.2 (3.17) | 3.1 (2.94) | 2.6 (3.10) | **.038** |
| Red Flag | 3.6 (2.93) | 5.3 (1.86) | 4.3 (2.66) | < **.001** |
| complete Treatment | 10.1 (6.97) | 7.7 (9.05) | 9.1 (8.01) | **.048** |
| Other | 2.2 (1.80) | 1.1 (1.61) | 1.7 (1.81) | < **.001** |
| **Total of utterances** | 38.9 (14.25) | 39.0 (16.06) | 39.0 (15.02) | .963 |

*Note.* Discharge communication characteristics and content across all patient-physician encounters and by study group at discharge, as rated by blind and independent raters.

[1] Linear ANOVA.

**3.2.2 Content.** Regarding the content of discharge communications, physicians conveyed on average 39.0 (*SD* = 15.0; range: 7 to 108) utterances during the discharge conversation with their patients (for visual representation of the raw data see S1 and S2 Figs). The S-trained physician group expressed significantly more utterances in the categories explicit structuring *(p <* .001), Advice on self-care *(p* = .038), Red flags *(p <* .001), and complete Treatment *(p* = .048) utterances, as compared to the E group. Physicians in the S group provided fewer diagnosis related *(p* = .012), and "other" *(p <* .001) utterances than physicians in the E group. Also, there were no significant differences regarding information on follow-up *(p* = .955) between the S-trained and the E-trained physician group, as well as the overall total of utterances *(p* = .963). Values are reported in Table 2.

## 3.3 Primary outcome: Immediate recall

An overview of patients' absolute and relative recall for all three time points of assessment are provided in Table 3 (for visual representation of the raw data see S1 and S2 Figs). It is important to note that patients' recall of information was conditional on the physicians mentioning of the respective information during discharge. Therefore, the number of patients in individual analyses varies largely because in some cases only few physicians mentioned a particular item. For instance, the book metaphor was used by 5 physicians in the E group (4.5%) while it was used by 73 physicians in the S group (85,9%). This result as well as the lower number of physicians mentioning advise on self-care and red-flag items (see Table 3) can be considered as additional manipulation checks, confirming the success of the manipulation of training content. Overall, immediate absolute recall of patients was on average 10.3 (*SD* = 4.7) utterances, corresponding to 27.8% of all utterances provided by the physicians.

The mixed-effects model for immediate recall had a substantial explanatory power ($R^2$ = .27). The model's intercept was at 3.62 (SE = 0.84, 95% CI [1.97, 5.27], *p* < .001). The effect of group was positive with a small, significant effect (beta = 1.31, SE = 0.58, std. beta = 0.28, *p* < .05). The effect of number of utterances given by the physician was positive with a significant medium effect (beta = 0.16, SE = 0.02, std. beta = 0.50, *p* < .001).

In summary, patients in the S-group had a significantly higher immediate recall in comparison to patients in the E-group immediately after discharge (see Table 3).

## 3.4 Secondary outcomes: Long-term recall

At day 7, patients recalled on average 6.5 (*SD* = 3.6) utterances. The mixed effects model for recall at 7 days had substantial total explanatory power ($R^2$ = .07). The model's intercept was at 3.99 (SE = 0.77, 95% CI [2.48, 5.50], *p* < .001). The effect of group was negative and can be considered as tiny and not significant (beta = -0.08, SE = 0.54, std. beta = -0.02, *p* = .886). The effect of number of utterances given by the physician was positive and can be considered as small and significant (beta = 0.06, SE = 0.02, std. beta = 0.27, *p* < .001).

At day 30, patients recalled on average 5.5 (*SD* = 3.0) utterances. The model had a substantial total explanatory power ($R^2$ = .04). The model's intercept was at 3.92 (SE = 0.77, 95% CI [2.41, 5.42], *p* < .001). The effect of group was positive and can be considered as very small and not significant (beta = 0.57, SE = 0.56, std. beta = 0.19, *p* = .310). The effect of number of utterances given by the physician was positive and can be considered as very small and not significant (beta = 0.03, SE = 0.02, std. beta = 0.16, *p* = .056).

In summary, there were no significant differences between the two groups regarding the 7 day recall performance. Also, we found no significant differences between the two groups at day 30 recall performance (see Table 3).

**Table 3. Recall of discharge information.**

| | Empathy | | Structure | | Total | | p-value[1] | |
|---|---|---|---|---|---|---|---|---|
| | **Absolute** | **Relative** | **Absolute** | **Relative** | **Absolute** | **Relative** | **Absolute** | **Relative** |
| **Immediately after discharge** | N = 111 | | N = 85 | | N = 196 | | | |
| Explicit structure (book metaphor) | | | | | | | .239 | .231 |
| *N-Miss* | 106 | 106 | 12 | 12 | 118 | 118 | | |
| *Mean (SD)* | 0.00 (0.00) | 0.00 (0.00) | 0.70 (1.31) | 0.15 (0.28) | 0.65 (1.28) | 0.14 (0.27) | | |
| *Range* | 0.00–0.00 | 0.00–0.00 | 0.00–5.00 | 0.00–1.00 | 0.00–5.00 | 0.00–1.00 | | |
| **In**formation on diagnosis | | | | | | | .486 | .081 |
| *N-Miss* | 1 | 1 | 0 | 0 | 1 | 1 | | |
| *Mean (SD)* | 3.94 (2.75) | 0.29 (0.19) | 3.67 (2.49) | 0.34 (0.20) | 3.82 (2.63) | 0.31 (0.20) | | |
| *Range* | 0.00–14.00 | 0.00–0.83 | 0.00–13.00 | 0.00–1.00 | 0.00–14.00 | 0.00–1.00 | | |
| **F**ollow-up | | | | | | | .834 | .219 |
| *N-Miss* | 10 | 10 | 2 | 2 | 12 | 12 | | |
| *Mean (SD)* | 2.30 (2.49) | 0.32 (0.29) | 2.23 (1.76) | 0.37 (0.31) | 2.27 (2.19) | 0.34 (0.30) | | |
| *Range* | 0.00–15.00 | 0.00–1.00 | 0.00–7.00 | 0.00–1.00 | 0.00–15.00 | 0.00–1.00 | | |
| **A**dvice on self-care | | | | | | | .124 | .064 |
| *N-Miss* | 52 | 52 | 16 | 16 | 68 | 68 | | |
| *Mean (SD)* | 0.78 (1.20) | 0.22 (0.33) | 1.14 (1.43) | 0.33 (0.37) | 0.98 (1.34) | 0.28 (0.35) | | |
| *Range* | 0.00–5.00 | 0.00–1.00 | 0.00–5.00 | 0.00–1.00 | 0.00–5.00 | 0.00–1.00 | | |
| **R**ed Flag | | | | | | | .718 | .776 |
| *N-Miss* | 32 | 32 | 1 | 1 | 33 | 33 | | |
| *Mean (SD)* | 1.37 (1.36) | 0.29 (0.31) | 1.45 (1.62) | 0.28 (0.30) | 1.41 (1.50) | 0.29 (0.30) | | |
| Range | 0.00–5.00 | 0.00–1.00 | 0.00–6.00 | 0.00–1.00 | 0.00–6.00 | 0.00–1.00 | | |
| **c**omplete **T**reatment | | | | | | | .968 | .394 |
| N-Miss | 6 | 6 | 6 | 6 | 12 | 12 | | |
| *Mean (SD)* | 2.28 (2.18) | 0.25 (0.22) | 2.29 (2.81) | 0.28 (0.27) | 2.28 (2.46) | 0.26 (0.24) | | |
| *Range* | 0.00–12.00 | 0.00–1.00 | 0.00–19.00 | 0.00–1.00 | 0.00–19.00 | 0.00–1.00 | | |
| Other | | | | | | | .627 | .570 |
| *N-Miss* | 20 | 20 | 40 | 40 | 60 | 60 | | |
| *Mean (SD)* | 0.20 (0.52) | 0.06 (0.19) | 0.16 (0.37) | 0.09 (0.24) | 0.18 (0.47) | 0.07 (0.20) | | |
| *Range* | 0.00–3.00 | 0.00–1.00 | 0.00–1.00 | 0.00–1.00 | 0.00–3.00 | 0.00–1.00 | | |
| **Recall performance** | | | | | | | **.049** | **.018** |
| *Mean (SD)* | 9.69 (4.78) | 0.26 (0.11) | 11.02 (4.52) | 0.30 (0.11) | 10.27 (4.70) | 0.28 (0.11) | | |
| *Range* | 1.00–30.00 | 0.04–0.55 | 1.00–30.00 | 0.09–0.64 | 3.00–26.00 | 0.04–0.64 | | |
| **7 days after discharge** | N = 98 | | N = 73 | | N = 171 | | | |
| Explicit structure (book metaphor) | | | | | | | .619 | .284 |
| *N-Miss* | 93 | 93 | 11 | 11 | 104 | 104 | | |
| *Mean (SD)* | 0.20 (0.45) | 0.20 (0.45) | 0.40 (0.90) | 0.09 (0.20) | 0.39 (0.87) | 0.10 (0.22) | | |
| *Range* | 0.00–1.00 | 0.00–1.00 | 0.00–4.00 | 0.00–1.00 | 0.00–4.00 | 0.00–1.00 | | |
| **In**formation on diagnosis | | | | | | | .188 | .599 |
| *N-Miss* | 1 | 1 | 0 | 0 | 1 | 1 | | |
| *Mean (SD)* | 2.98 (2.53) | 0.21 (0.18) | 2.51 (1.97) | 0.23 (0.17) | 2.78 (2.31) | 0.22 (0.17) | | |
| *Range* | 0.00–10.00 | 0.00–0.83 | 0.00–11.00 | 0.00–0.69 | 0.00–11.00 | 0.00–0.83 | | |
| **F**ollow-up | | | | | | | .501 | .935 |
| *N-Miss* | 8 | 8 | 1 | 1 | 9 | 9 | | |
| *Mean (SD)* | 1.14 (1.46) | 0.17 (0.23) | 1.00 (1.21) | 0.17 (0.22) | 1.08 (1.35) | 0.17 (0.22) | | |
| *Range* | 0.00–6.00 | 0.00–1.00 | 0.00–5.00 | 0.00–1.00 | 0.00–6.00 | 0.00–1.00 | | |
| **A**dvice on self-care | | | | | | | .454 | .863 |

*(Continued)*

**Table 3.** (Continued)

| | Empathy | | Structure | | Total | | p-value[1] | |
|---|---|---|---|---|---|---|---|---|
| | **Absolute** | **Relative** | **Absolute** | **Relative** | **Absolute** | **Relative** | **Absolute** | **Relative** |
| *N-Miss* | 48 | 48 | 13 | 13 | 61 | 61 | | |
| *Mean (SD)* | 0.78 (1.33) | 0.14 (0.22) | 0.60 (1.18) | 0.15 (0.26) | 0.68 (1.25) | 0.14 (0.24) | | |
| *Range* | 0.00–5.00 | 0.00–0.71 | 0.00–7.00 | 0.00–1.00 | 0.00–7.00 | 0.00–1.00 | | |
| **Red Flag** | | | | | | | .062 | .085 |
| *N-Miss* | 28 | 28 | 1 | 1 | 29 | 29 | | |
| *Mean (SD)* | 0.63 (1.14) | 0.12 (0.23) | 1.03 (1.37) | 0.19 (0.25) | 0.83 (1.28) | 0.15 (0.24) | | |
| *Range* | 0.00–4.00 | 0.00–1.00 | 0.00–5.00 | 0.00–1.00 | 0.00–5.00 | 0.00–1.00 | | |
| **complete Treatment** | | | | | | | .115 | .534 |
| *N-Miss* | 6 | 6 | 6 | 6 | 12 | 12 | | |
| *Mean (SD)* | 1.61 (1.90) | 0.18 (0.20) | 1.16 (1.50) | 0.16 (0.19) | 1.42 (1.76) | 0.17 (0.20) | | |
| *Range* | 0.00–13.00 | 0.00–1.00 | 0.00–9.00 | 0.00–1.00 | 0.00–13.00 | 0.00–1.00 | | |
| Other | | | | | | | .180 | .211 |
| *N-Miss* | 15 | 15 | 34 | 34 | 49 | 49 | | |
| *Mean (SD)* | 0.12 (0.42) | 0.05 (0.17) | 0.03 (0.16) | 0.01 (0.08) | 0.09 (0.36) | 0.04 (0.15) | | |
| *Range* | 0.00–3.00 | 0.00–1.00 | 0.00–1.00 | 0.00–0.50 | 0.00–3.00 | 0.00–1.00 | | |
| Recall performance | | | | | | | .936 | .894 |
| *Mean (SD)* | 6.47 (3.83) | 0.18 (0.10) | 6.42 (3.30) | 0.17 (0.11) | 6.45 (3.60) | 0.18 (0.11) | | |
| *Range* | 0.00–18.00 | 0.00–0.53 | 0.00–14.00 | 0.00–0.57 | 0.00–18.00 | 0.00–0.57 | | |
| **30 days after discharge** | N = 83 | | N = 60 | | N = 143 | | | |
| Explicit structure (book metaphor) | | | | | | | .324 | .249 |
| *N-Miss* | 78 | 78 | 9 | 9 | 87 | 87 | | |
| *Mean (SD)* | 0.00 (0.00) | 0.00 (0.00) | 0.39 (0.87) | 0.09 (0.18) | 0.36 (0.84) | 0.08 (0.18) | | |
| *Range* | 0.00–0.00 | 0.00–0.00 | 0.00–3.00 | 0.00–0.60 | 0.00–3.00 | 0.00–0.60 | | |
| **Information on diagnosis** | | | | | | | .722 | .188 |
| *N-Miss* | 1 | 1 | 0 | 0 | 1 | 1 | | |
| *Mean (SD)* | 2.44 (2.09) | 0.18 (0.15) | 2.32 (1.93) | 0.21 (0.17) | 2.39 (2.01) | 0.19 (0.16) | | |
| *Range* | 0.00–7.00 | 0.00–0.55 | 0.00–8.00 | 0.00–0.50 | 0.00–8.00 | 0.00–0.55 | | |
| Follow-up | | | | | | | .573 | .678 |
| *N-Miss* | 8 | 8 | 1 | 1 | 9 | 9 | | |
| *Mean (SD)* | 0.85 (1.20) | 0.14 (0.21) | 0.97 (1.07) | 0.15 (0.17) | 0.90 (1.14) | 0.14 (0.19) | | |
| *Range* | 0.00–4.00 | 0.00–1.00 | 0.00–3.00 | 0.00–0.67 | 0.00–4.00 | 0.00–1.00 | | |
| **Advice on self-care** | | | | | | | .618 | .835 |
| *N-Miss* | 39 | 39 | 12 | 12 | 51 | 51 | | |
| *Mean (SD)* | 0.68 (1.03) | 0.14 (0.22) | 0.56 (1.24) | 0.15 (0.28) | 0.62 (1.14) | 0.14 (0.25) | | |
| *Range* | 0.00–4.00 | 0.00–1.00 | 0.00–7.00 | 0.00–1.00 | 0.00–7.00 | 0.00–1.00 | | |
| **Red Flag** | | | | | | | .692 | .746 |
| *N-Miss* | 25 | 25 | 1 | 1 | 26 | 26 | | |
| *Mean (SD)* | 0.84 (1.14) | 0.16 (0.25) | 0.93 (1.24) | 0.18 (0.24) | 0.89 (1.19) | 0.17 (0.24) | | |
| *Range* | 0.00–3.00 | 0.00–1.00 | 0.00–4.00 | 0.00–0.80 | 0.00–4.00 | 0.00–1.00 | | |
| **complete Treatment** | | | | | | | .134 | .757 |
| *N-Miss* | 4 | 4 | 6 | 6 | 10 | 10 | | |
| *Mean (SD)* | 1.11 (1.34) | 0.12 (0.18) | 0.80 (0.94) | 0.13 (0.21) | 0.98 (1.20) | 0.13 (0.19) | | |
| *Range* | 0.00–5.00 | 0.00–1.00 | 0.00–3.00 | 0.00–1.00 | 0.00–5.00 | 0.00–1.00 | | |
| Other | | | | | | | .301 | .848 |
| *N-Miss* | 11 | 11 | 27 | 27 | 38 | 38 | | |
| *Mean (SD)* | 0.11 (0.43) | 0.04 (0.15) | 0.03 (0.17) | 0.03 (0.17) | 0.09 (0.37) | 0.03 (0.16) | | |

*(Continued)*

**Table 3.** (Continued)

| | Empathy | | Structure | | Total | | p-value[1] | |
|---|---|---|---|---|---|---|---|---|
| | Absolute | Relative | Absolute | Relative | Absolute | Relative | Absolute | Relative |
| *Range* | 0.00–3.00 | 0.00–1.00 | 0.00–1.00 | 0.00–1.00 | 0.00–3.00 | 0.00–1.00 | | |
| Recall performance | | | | | | | .436 | .330 |
| *Mean (SD)* | 5.30 (3.06) | 0.15 (0.09) | 5.70 (2.94) | 0.16 (0.09) | 5.47 (3.01) | 0.15 (0.09) | | |
| *Range* | 0.00–12.00 | 0.00–0.38 | 0.00–14.00 | 0.00–0.44 | 0.00–14.00 | 0.00–0.44 | | |

*Note*. Recall of discharge information immediately after discharge, one week and one months later, and adherence to recommendations one week and one months later. Recall of discharge information is reported conditional on the respective information being mentioned during discharge. Accordingly, missing values can be explained by the physician not mentioning the respective information during discharge, which reduces the number of patients in the respective analysis. No imputation of missing data was conducted. *Absolute*, absolute value; *relative*, relative value; *SD*, standard deviation; *range*, the area of variation between upper and lower limits; *N-miss*, number of missing values.

[1] Linear ANOVA.

### 3.5 Secondary outcomes: Feasibility, adherence, satisfaction and patient-physician-relationship

*Feasibility* was shown by a 69.6% training completion rate. Out of 117 eligible physicians 80 completed all teaching modules and 63 physicians discharged patients under study conditions.

*Patients adhered* to an average of a total of 1.5 recommendations given by the physicians ($SD = 1.2$) after 30-days (Table 4). Adherence to the recommendations given by the physician within 30 days was significantly higher in the S-group as compared to the E-group for absolute ($p = .002$) and relative ($p = .004$) values. In addition, adherence to recommendations within a month correlated significantly with immediate recall in the E group ($r(81) = .29, p = .007$) and the S group ($r(58) = .32, p = .012$).

Patients' ratings were high in both groups on all four dimensions of *satisfaction* (Table 4). Patients in the S-group had higher satisfaction ratings on structure ($p = .028$) and recommendation ($p = .021$) compared with the E group, and groups did not differ on their ratings of comprehension ($p = .209$) and informativeness ($p = .175$).

There were no significant differences between the groups on the subscales of the *perceived quality of relationship between patient and physician* measured with the PRA-D (Table 4).

## 4 Discussion

We designed two communication trainings for physicians, involving either explicit information structuring or empathy skills. We evaluated the feasibility of conducting these communication trainings within the ED of the University Hospital in Basel, Switzerland and we evaluated the respective effects of the two trainings on patients' recall of discharge information, their adherence to recommendations during a 30-day period after discharge, and their satisfaction with the physician after discharge from the ED. The main finding of our study shows superiority of S over E on patients' immediate recall performance. Further, the feasibility of trainings containing three short sessions in the first weeks of residency was shown, and patients in the S group showed higher adherence to recommendations and were similarly satisfied with the physician compared with the E group.

Empathy skills have been shown to improve patients' outcomes and satisfaction [36]. Our manipulation check revealed that empathic communication skills were shown similarly in both training groups. Thus, our study evaluated the additional effect of focusing on structuring skills as a complement to communicating in an empathic way. This ensured that the

**Table 4. Patients' adherence, satisfaction, and PRA-D scores.**

| | Empathy | Structure | Total | p-value[1] |
|---|---|---|---|---|
| | (N = 111) | (N = 85) | (N = 196) | |
| **Adherence to recommendations** | | | | |
| Recommendations given by the physician | | | | .205 |
| *Mean (SD)* | 4.38 (2.21) | 4.78 (1.81) | 4.55 (2.05) | |
| *Range* | 0.00–10.00 | 2.00–12.00 | 0.00–12.00 | |
| Adherence within one month (Absolute) | | | | **.002** |
| *N-Miss* | 28 | 25 | 53 | |
| *Mean (SD)* | 1.19 (1.11) | 1.82 (1.27) | 1.45 (1.21) | |
| *Range* | 0.00–5.00 | 0.00–5.00 | 0.00–5.00 | |
| Adherence within one month (Relative) | | | | **.004** |
| *N-Miss* | 29 | 25 | 54 | |
| *Mean (SD)* | 0.27 (0.25) | 0.39 (0.27) | 0.32 (0.26) | |
| *Range* | 0.00–1.00 | 0.00–1.00 | 0.00–1.00 | |
| **Patient satisfaction** | | | | |
| Comprehensibility of the discharge communication | | | | .209 |
| *N-Miss* | 0 | 4 | 4 | |
| *Median (IQR)* | 9.80 (1.05) | 9.90 (1.0) | 9.80 (1.00) | |
| Structuredness of the discharge communication | | | | **.028** |
| *N-Miss* | 0 | 4 | 4 | |
| *Median (IQR)* | 9.10 (2.00) | 9.40 (1.30) | 9.15 (2.00) | |
| Recommendation of the physician to family and friends | | | | **.021** |
| *N-Miss* | 0 | 4 | 4 | |
| *Median (IQR)* | 9.50 (1.00) | 10.00 (1.00) | 10.00 (1.00) | |
| Informativeness of the discharge communication | | | | .175 |
| *N-Miss* | 6 | 11 | 17 | |
| *Median (IQR)s* | 9.40 (2.00) | 9.50 (1.00) | 9.50 (1.5) | |
| **Patient Reactions Assessment (PRA-D)** | | | | |
| Patient affective index | | | | .199 |
| *N-Miss* | 12 | 11 | 23 | |
| *Mean (SD)* | 30.21 (4.66) | 31.08 (3.99) | 30.58 (4.40) | |
| Patient communication index | | | | .305 |
| *N-Miss* | 12 | 11 | 23 | |
| *Mean (SD)* | 30.75 (5.73) | 31.58 (4.59) | 31.10 (5.27) | |
| Patient information index | | | | .200 |
| *N-Miss* | 12 | 11 | 23 | |
| *Mean (SD)* | 27.44 (5.37) | 28.50 (5.31) | 27.90 (5.35) | |
| Total | | | | .124 |
| *N-Miss* | 12 | 11 | 23 | |
| *Mean (SD)* | 88.40 (11.88) | 91.16 (11.23) | 89.58 (11.65) | |

*Note.* Patients' adherence to physicians' recommendations, self-rated satisfaction, and patient-physician-relationship, across all patient-physician encounters and by study group. Missing satisfaction and PRA-D values can be explained by the fact that the respective items were added to the questionnaire later on in the process of data collection, after the first patients had been included in the study; the pattern can be considered completely at random accordingly, and no imputation of missing values was conducted. *N-miss*, number of missing values. [1] Linear ANOVA

comparison group was not obviously inferior to the explicit structure intervention. Therefore, the comparison of S versus E regarding recall was expected to be of only small superiority in our study. The patients who were treated by a physician who used the trained structuring

methods recalled on average 1.33 more information items compared with patients who were treated by physicians who were trained in empathy. This observed small difference between the two training groups are in line with the findings of a meta-analysis on the effects of communication training on patient outcomes [37] in oncology. The difference of more than one item additionally recalled in the S group was judged to be relevant by most emergency physicians (data not shown), although a limit for clinical relevance has never been formally established. The relevance of the superior recall observed in patients in the S group is underlined by the fact that the enhanced recall was associated with enhanced adherence to the physicians' recommendations in the S group as compared with patients in the E group. Given the rather low overall adherence to recommendations even a small increase in adherence may be considered a success. Previous findings confirm our observation that patient recall is associated with adherence to recommendations [6, 38, 39].

In the present study, patients' satisfaction was high in both groups, but patients in the S group more strongly endorsed they would recommend their physician to family and friends in comparison to physicians in the E group. This finding was unexpected, as we would have expected patient satisfaction to be higher in the E group. However, the similar results between the two groups on satisfaction might reflect the observation that discharge communications did not differ with respect to empathy between the two groups, as rated by independent and blind raters. Overall, our results show that the non-information outcomes were achieved equally well between the empathy and the information intervention, meaning patients value the quality of their information from healthcare providers as well as the quality of their relationships with providers.

Our study implemented parsimonious trainings, which were shorter than previously studied interventions [37, 40], but lead to a comparably high number of physicians completing the training (69.6%). This demonstrates the feasibility of applying brief communication trainings within ED practice. Nevertheless, the minimal effort and duration of trainings may be one reason for the observed small differences between S and E groups. However, the short duration of trainings match with the requirements within an extremely busy ED. Taken together, a communication training that is highly standardized, but of minimal duration, can be considered as feasible, is associated with a high satisfaction, and, if specifically focusing on information structuring, may be superior to a training focusing on empathic communication alone, with respect to patients' information recall and longer-term adherence to recommendations.

Our study has several limitations. First, our study was a single center study, the number of physicians in each cluster was small and the patient sample comprised patients with chest and abdominal pain only. However, our population is comparable to other European EDs and our sample of physicians is representative. All but two eligible physicians were initially included, and 69.6% of the physicians completed all three modules of the communication training. Patients with chest and abdominal pain were chosen as a study population because these complaints have a high prevalence in Eds [41, 42]. Second, the communication trainers could not be blinded to the interventions. Yet, physicians' test scores in standardized tests showed that both groups were similarly successful in learning and applying their communication skills. Third, the current study did not check for relevance of the recalled utterances because experts cannot easily define which items are most relevant in discharge communication [43]. Fourth, patients' adherence was based on self-reports. We were not able to check whether patients actually adhered to the recommendations (e.g. by testing serum concentrations of the prescribed drugs). However, patients' self-reported adherence was low, and bias would likely apply to both groups and should therefore not account for differences between E and S trainings. Fifth, 46.0% of the randomized patients were not available for analyses at last follow-up. While in a more standardized setting this number would be considered high, it is important to

note, that half of the losses to follow up did not even take part in the immediate assessments. Of those patients who were assessed immediately after discharge 73.0% were assessed one month later again, which reflects a reasonable completion rate. Sixth, we did not include a standard care or no-training control group. For future research it would be interesting to investigate the effects of both presented trainings with the standard of care in EDs, in order to clarify the absolute efficacy of implementing such trainings. It is also possible that focusing on empathy may even interfere with the physicians' primary tasks of conveying relevant information, or with patients' information recall, as previous research may suggest [44]. However, our study did not aim at investigating this question. Seventh, though training helped with immediate recall, the waning effect remains to be a concern. We therefore suggest a second structured information as a follow-up to boost recall and support patient adherence. Effects of such an additional structured intervention, however, should be studied in the future. Last, the original clinical trial protocol that was submitted to the local ethics committee was amended: 1) we were planning to include a total of 400 participants within one year to guarantee a power of over .9. Including 400 patients within one year was not feasible as many of the screened patients did not meet the inclusion criteria. This led to an extension of the study period that was approved by the local ethics committee, an increase in the number of physician clusters and a decrease of included patients (i.e. 196). Nevertheless, a smaller sample size of 200 patients still guaranteed satisfactory power level of above .8 and the longer recruitment did not bear the risk of biasing the study results. 2) feasibility was added as a secondary outcome although this variable was not included in the originally published protocol. This was important as the communication skills training under investigation needed to be easy to follow and implementable in clinical practice.

## 5 Conclusion

Our results suggest that teaching explicit information structuring to physicians is feasible in terms of physicians' training completion rate, and the application of their skills in practice. Our study suggests that the mix of empathic communication and structured information provision may improve patients' recall of information, their adherence to recommendations and also their satisfaction with ED treatment. Given the low intensity of training, and the design of the study as a comparative trial, only a small superiority of information structuring could be shown in terms of patients' immediate recall and adherence to recommendations. Finally, patient satisfaction was high and there was no trade-off in the information structuring group as compared to the group with physicians trained in empathy skills. The observed findings, but also the lack of a standard of care or no-training control in our study, as well as the observation of inconsistent findings in previous studies on implementing communication strategies in ED to increase information patients' recall and adherence to recommendations indicate the need for further research.

## Supporting information

**S1 Checklist.**
(DOC)

**S1 File. Content of the communication training.**
(DOCX)

**S1 Protocol.**
(DOCX)

**S1 Table. Coding scheme.**
(DOCX)

**S1 Fig. Visual representation of the raw data.**
(DOCX)

**S2 Fig. Visual representation of the raw data from discharge to 30 days follow-up assessment.**
(DOCX)

## Acknowledgments

The authors thank Marianne Benz, Leona Knüsel, Dominik Maiori, Valentina Steffen, Doreen Eckardt, Laura Fässler, Anna Becker, David Bachmann, Galya Iseli, and Isabelle Keller for their support in data collection and Laura Wiles for editing the manuscript.

## Author Contributions

**Conceptualization:** Victoria Siegrist, Rui Mata, Wolf Langewitz, Ralph Hertwig, Roland Bingisser.

**Data curation:** Victoria Siegrist, Stephan Furger.

**Formal analysis:** Victoria Siegrist, Stephan Furger.

**Funding acquisition:** Roland Bingisser.

**Investigation:** Victoria Siegrist.

**Methodology:** Victoria Siegrist, Wolf Langewitz, Roland Bingisser.

**Project administration:** Victoria Siegrist.

**Resources:** Roland Bingisser.

**Software:** Victoria Siegrist.

**Supervision:** Rui Mata, Wolf Langewitz, Ralph Hertwig, Roland Bingisser.

**Validation:** Victoria Siegrist, Rui Mata, Wolf Langewitz, Heike Gerger.

**Visualization:** Victoria Siegrist.

**Writing – original draft:** Victoria Siegrist.

**Writing – review & editing:** Victoria Siegrist, Rui Mata, Wolf Langewitz, Heike Gerger, Stephan Furger, Ralph Hertwig, Roland Bingisser.

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
