## [Decision Letter · Decision Letter 0]

4 Dec 2020

PONE-D-20-21645

Does Information Structuring Improve Recall of Discharge Information? A Cluster Randomized Clinical Trial

PLOS ONE

Dear Dr. Siegrist,

Thank you for submitting your manuscript to PLOS ONE. After careful consideration, we feel that it has merit but does not fully meet PLOS ONE’s publication criteria as it currently stands. Therefore, we invite you to submit a revised version of the manuscript that addresses the points raised during the review process.

The manuscript has been evaluated by three reviewers, and their comments are available below.

The reviewers have raised a number of concerns that need attention. They request additional information on methodological aspects of the study (such as additional information on sample size determination and statistical analyses) and a deeper synthesis and discussion of your results to support your conclusions.

Could you please revise the manuscript to carefully address the concerns raised?

We look forward to receiving your revised manuscript.

Kind regards,

Beryne Odeny

Staff Editor

PLOS ONE

Journal Requirements:

2. Thank you for including your ethics statement:  "The study was approved in writing by the local ethics committee (EKNZ 2014-379) and the protocol was published on ClinicalTrials.gov (NCT02468869)."   

4. Please upload a copy of Figure 3, to which you refer in your text (line 273). If the figure is no longer to be included as part of the submission please remove all reference to it within the text.

5. Please include a caption for figure 3.

Reviewers' comments:

Reviewer's Responses to Questions

**Comments to the Author**

1. Is the manuscript technically sound, and do the data support the conclusions?

Reviewer #1: Yes

Reviewer #2: Partly

Reviewer #3: Yes

2. Has the statistical analysis been performed appropriately and rigorously? 

Reviewer #1: Yes

Reviewer #2: No

Reviewer #3: Yes

3. Have the authors made all data underlying the findings in their manuscript fully available?

Reviewer #1: Yes

Reviewer #2: Yes

Reviewer #3: Yes

4. Is the manuscript presented in an intelligible fashion and written in standard English?

Reviewer #1: Yes

Reviewer #2: Yes

Reviewer #3: Yes

5. Review Comments to the Author

Reviewer #1: This is a very well-done study of an oft discussed but rarely studied phenomenon. While the ultimate conclusions of the research are not earth-shattering, they represent a solid step forward in taking communication processes seriously in health settings. I have a few observations that might help the authors to improve their presentation and arguments.

1. I noticed early on that the sample was restricted to residents and that you say in passing that the residents conducted the patient discharge. In many parts of the world the discharge conversation is delegated or assigned to nurses or medical social workers, so it would be important for you to comment on whether residents always do discharges at this hospital or whether there is some variability…and whether you think your findings regarding structured communication would apply if other roles were involved.

2. I found it fascinating that in your manipulation check you discovered that both groups of residents were equally empathic; I suppose that speaks well for their training in general. This does mean that your finding of the value of structured communication really is on top of a baseline level of empathy. You make this clear in the conclusion, but I just wanted to comment that from your hypotheses I was anticipating that the empathy training was not going to turn out to be very useful.

3. I could use a little more detail on how you measured immediate recall, especially since it is one of the few places you found a significant difference. I went into the study protocol to see that there were five-minute conversations, but there ought to be more about this in the paper.

4. I suppose that the most significant finding is that structured communication drives adherence weeks out, so it is a little disappointing that your measure of this is solely self-report. But I see no reason to think that these self-reports would vary for other reasons that would counteract your conclusions. An interesting follow-up study might identify more behavioral outcome measures to really clinch the value of this approach.

5. I think it might help for you to explain that structured communication can take many forms and that you chose a particular operationalization with INFARCT; I know you want to make claims about the structured approach but there could be some different ways to get to the same end.

6. Finally, as a qualitative health researcher myself I found myself wanting a closer analysis of these discharge conversations that goes beyond the coding and capturing of utterances. Mere recall of an utterance is one thing, but there are a number of other communication factors that could also improve the likelihood of patients grasping the physician’s meaning. This is not a criticism of your study but an observation that other methods such as conversation analysis or ethnography could shed even more light on this phenomenon.

Reviewer #2: The manuscript entitled ‘Does Information Structuring Improve Recall of Discharge Information? A Cluster Randomized Clinical Trial.’ with the aim to assess the information structuring using the book metaphor and the InFARcT mnemonic and improvement on patients’ information recall, their adherence to instructions and patients’ satisfaction.

This study is quite interesting, however, the manuscript requires improvement.

Comments

Abstract

Line 34, one or two lines of introduction/background to be provided before the objectives.

Methods

The duration of the patients in ED and assessment to be clearly stated.

Line 123-127, the allocation concealment, blinding to be stated.

Line 146-149, the period of assessment to be clearly stated.

Line 160, for Immediately discharged assessment, the period/time point to be clearly stated.

Line 185 - 190, for the telephone conversation recording at 7 and 30 days and the method of recording to be clearly stated.

Sample size calculation and statistical analysis

1 or 2 tailed test, sample size for each group and attrition rates consideration to be stated. The accepted level of significance to be stated.

Results

Line 256, what range refers to, to be clearly stated.

Table 1, at least one decimal point for percentages figures. Symbol <= , >= to be replaced with ≤, ≥ respectively. The decimal points for p values to be standardized. All the decimal points for the figures quoted in text to follow the exact figures in the table.

However, based on CONSORT guidelines, all statistical tests on group comparison at baseline to be avoided. As such all the description on the ‘statistical analysis’ on baseline to be avoided.

Line 287 3.2.1., Line 302 3.2.2, Line 314 Table 2, the time point/period of assessment to be clearly stated.

Line 332-337 & 342-355, results to be presented/tabulated in table form.

Line 334, 350, for substantial total explanatory power values, need to discuss more on the values and the reasons.

Line 378-379, the p value for t test or correlation test to clearly separated.

Line 383, for informativeness (p = .199), p value is 0.175. 0.199 is for patient affective index.

Line 385, p >0.124 to follow Table 4 p=0.124

Table 2, statistical test to be denoted in the table footnote.

Table 3, Absolute, Relative, Range to be clearly denoted in the table footnote.

Table 3 & 4, N- Miss to be clearly denoted in the table footnote or replaced with the word ‘missing’.

Effect size could be provided where applicable.

A section to be provided to describe the missing data at overall and various time period i.e percentages etc.

For the list of references, journal name and style to follow journal format.

Reviewer #3: This is an interesting study that tested the effectiveness of the structured discharge information presentation training relative to empathy training. The study was well conducted and the paper well written. The findings should be of interest to researchers and clinicians who are interested in intervention aimed to improve clinician-patient communication.

I have a few observations and minor suggestions for revisions.

1. There is no reason to think INFARcT training would be superior to empathy training on any outcomes other than recall and structure of information giving. The fact that information structure was superior to empathy was largely because the coding scheme was based on the INFARcT training itself. This is less about quality of information giving and more about fidelity of the intervention. I think this should be acknowledged in the discussion.

2. While there were differences in self reported adherence and recommending the physician, there is not theoretical reason provided for why this should be the case. While the study did not test a theoretical explanation, the authors should offer at least some speculation because I feel it inadequate to draw any conclusions on adherence (even if self-reported) and physician recommendation simply based on the assumption 'it must have been INFARcT training.'

3. I would argue that the rationale for comparing INFARcT to empathy training is that empathy is considered the gold standard for effective communication is not convincing. Empathy is important, but quality of information exchange, perhaps shared decision-making (though less relevant in ED) are just as important. Thus, a better rationale needs to be provided (perhaps authors were interested in comparing an information intervention to a relationship-focused one). Or argue empathy as a control condition because it does not focus on information per se.

4. I think the physician-patient relationship outcome is mislabeled. The PRA assesses patient perceptions of the quality of communication--physician informative, caring, and patient feels comfortable communicating. I suggest label it perceived quality of communication or perceived quality of relationship (i.e., needs 'quality' in the label).

5. In discussion, I think authors could say that the non information outcomes were acheived equally (for the most part) between the empathy and the information intervention, meaning patients value the quality of their information from providers as well as the quality of their relationships with providers.

6. PLOS authors have the option to publish the peer review history of their article (what does this mean?). If published, this will include your full peer review and any attached files.

Reviewer #1: No

Reviewer #2: No

Reviewer #3: No

---

## [Author Response · Author response to Decision Letter 0]

31 May 2021

Dear editor, dear reviewers, 

We would like to express our thanks for your valuable inputs and believe that the manuscript has improved by your comments. 

Point-to-point reply to editor’s comments:

Answer: We have thoroughly revised our manuscript according to PLOS ONE’s style requirements. 

2. Thank you for including your ethics statement: "The study was approved in writing by the local ethics committee (EKNZ 2014-379) and the protocol was published on ClinicalTrials.gov (NCT02468869)." 

Answer: We have amended the current ethics statement including the full name of the ethics committee that approved our study. You can find the full name of the ethics committee that approved the study on page 5-6, lines 123-128. Please find the updated wording below.

“The study was approved by the local ethics committee “Ethikkommission Nordwest- und Zentralschweiz” (EKNZ 2014-379) on December 3, 2014 and the protocol was published on ClinicalTrials.gov (NCT02468869).”

Answer: We have added the current ethics statement including the full name of the ethics committee that approved our study to the Ethics Statement field of the submission

Answer: After consultation with the responsible ethics committee, there are no restrictions as long as data is anonymized. Therefore, we uploaded the minimal anonymized data set necessary to replicate our study findings to a stable, public repository and provide you with the relevant URL: https://osf.io/84q3r/

5. Please upload a copy of Figure 3, to which you refer in your text (line 273). If the figure is no longer to be included as part of the submission please remove all reference to it within the text.

 Answer: We have updated the Figure name from Figure 3 to Figure 2, that was referred to in line 273 (now, it is line: 381).

6. Please include a caption for figure 3.

 Answer: A caption for Figure 2 is already included (no Figure 3 will be shown in the manuscript).

Answer: We now have included captions for our supporting information files at the end of the manuscript on page 35 and updated any in-text citations to match accordingly.

Point-to-point reply to reviewers’ comments:

Reviewer #1

This is a very well-done study of an oft discussed but rarely studied phenomenon. While the ultimate conclusions of the research are not earth-shattering, they represent a solid step forward in taking communication processes seriously in health settings. I have a few observations that might help the authors to improve their presentation and arguments.

1. I noticed early on that the sample was restricted to residents and that you say in passing that the residents conducted the patient discharge. In many parts of the world the discharge conversation is delegated or assigned to nurses or medical social workers, so it would be important for you to comment on whether residents always do discharges at this hospital or whether there is some variability…and whether you think your findings regarding structured communication would apply if other roles were involved.

Answer: Thank you for this input. We were indeed not aware of health care systems in which nurses or even social workers discharge patients from an Emergency Department. Therefore, we have not tested for this option, but we believe that similar outcomes might be found in other professional groups as well: Why should the effect on patients’ recall differ according to the information-giving subject’s profession? As we pointed out, the approach is minimalistic, teaching is highly formalized and concise, and the results, as shown by the plots in the Appendix section “S3 Visual representation of the raw data”, indicate that the formal teaching of structuring discharge information actually resulted in a higher structure of information-giving. 

2. I found it fascinating that in your manipulation check you discovered that both groups of residents were equally empathic; I suppose that speaks well for their training in general. This does mean that your finding of the value of structured communication really is on top of a baseline level of empathy. You make this clear in the conclusion, but I just wanted to comment that from your hypotheses I was anticipating that the empathy training was not going to turn out to be very useful.

Answer: Indeed, we were fascinated by the same finding as most residents tend to have a short professional experience, suffer from extreme pressure due to high workload, and have a high turn-over in most European Emergency Departments. We are not aware of any study with such high empathy ratings in Emergency Medicine. We have updated the wording to make this point even clearer in the discussion. Please find the updated wording below (page 26, lines 542-546).

“Our manipulation check revealed that empathic communication skills were shown similarly in both training groups. Thus, our study evaluated the additional effect of focusing on structuring skills as a complement to communicating in an empathic way. This ensured that the comparison group was not obviously inferior to the explicit structure intervention.”

3. I could use a little more detail on how you measured immediate recall, especially since it is one of the few places you found a significant difference. I went into the study protocol to see that there were five-minute conversations, but there ought to be more about this in the paper.

Answer: We have added a more in-depth description of the measurement of immediate recall in the manuscript under “2.4.1 Primary outcome: Immediate discharge information recall” (page 8, lines 193-198):

“Recall was assessed by asking patients the question: “Please share with me all the information that you recall from your discharge communication.” After the last utterance was shared proactively by the patient, the interviewer would probe and ask “is there any additional information that you remember from your discharge communication?”. The interview was stopped once the patient stated that there is nothing else that he or she recalls.”

4. I suppose that the most significant finding is that structured communication drives adherence weeks out, so it is a little disappointing that your measure of this is solely self-report. But I see no reason to think that these self-reports would vary for other reasons that would counteract your conclusions. An interesting follow-up study might identify more behavioral outcome measures to really clinch the value of this approach.

Answer: As the study design was quite ambitious for a communication study, particularly in the challenging environment of an inner-city Emergency Department, it was not feasible to use a more “objective measurement” for adherence. A large part of adherence studies relies on self-report. Objective measurements of, e.g. compliance with medication, is extremely challenging. Adherence to asthma medication was shown to be similar, if self-reports were compared to measurement of the contents of spray containers. However, micro-chip packed containers used in other studies showed that the consumption may occur very asymmetrically, such as in the last few hours before the study visit. Therefore, it would have been a challenge to find an “objective method” to measure adherence in 200 patients, not only with individual medication, but also with other medical instructions to be followed. As the method was exactly the same for both groups, we believe that even if the results may be biased in favor of adherence, it is impacted in both groups in a similar way. We agree that more studies are needed in this respect. 

5. I think it might help for you to explain that structured communication can take many forms and that you chose a particular operationalization with INFARCT; I know you want to make claims about the structured approach but there could be some different ways to get to the same end.

Answer: This is a point well taken. This study uses a “proof-of-principle” approach. For decades, the theory that formally structured information-giving could aid recall in critical situations, such as in emergency discharge, was widely discussed and was part of teaching in medical communication. However, there was not one real-life study to gather evidence on this topic. We have added the following sentence in the introduction of the manuscript to clarify that structured communication can take many forms (page 5, lines: 100-102):

“Structuring the content of information given can take many forms; in pre-medication visits (22) the topics to be dealt with differ from discharge communication from patient to patient.”

6. Finally, as a qualitative health researcher myself I found myself wanting a closer analysis of these discharge conversations that goes beyond the coding and capturing of utterances. Mere recall of an utterance is one thing, but there are a number of other communication factors that could also improve the likelihood of patients grasping the physician’s meaning. This is not a criticism of your study but an observation that other methods such as conversation analysis or ethnography could shed even more light on this phenomenon.

Answer: We absolutely agree that there is much more to this very important aspect in medicine and more specifically communication in healthcare. As discharge communication is closely linked to patient satisfaction and medical outcomes, it needs to be studied in much more detail. Other methods, such as conversation analysis or ethnography could indeed shed much more light on this phenomenon. On the other hand, a quantitative proof-of-principle approach cannot be replaced by qualitative health research – neither can qualitative research be replaced by the assessment of quantitative effects in larger populations. We therefore believe that this research is a good starting point for future research.

Reviewer #2

The manuscript entitled ‘Does Information Structuring Improve Recall of Discharge Information? A Cluster Randomized Clinical Trial.’ with the aim to assess the information structuring using the book metaphor and the InFARcT mnemonic and improvement on patients’ information recall, their adherence to instructions and patients’ satisfaction.

This study is quite interesting, however, the manuscript requires improvement.

Comments

Abstract

Line 34, one or two lines of introduction/background to be provided before the objectives.

Answer: Thank you for this important input. We have added more background information to the abstract in the objectives section. Please find the updated wording below (page 2, lines: 36-42):

“The impact of the quality of discharge communication between physicians and their patients is critical on patients’ health outcomes. Nevertheless, low recall of information given to patients at discharge from emergency departments (EDs) is a well-documented problem. Therefore, we investigated the outcomes and related benefits of two different communication strategies: Physicians were instructed to either use empathy (E) or information structuring (S) skills hypothesizing superior recall by patients in the S group.”

Methods

The duration of the patients in ED and assessment to be clearly stated.

Answer: We agree with the reviewer that it would have been interesting to have a look at the duration of the patients in ED and the assessments but given the complexity of the study design and the focus on the effect of structured vs. unstructured discharge communication, these measures were not assessed.

Line 123-127, the allocation concealment, blinding to be stated.

Answer: We have added information on the allocation concealment. Please find the updated wording below (page 6, lines 133-138):

“As study physicians we included new residents starting at the ED of the University Hospital Basel. Physicians were clustered according to their first day at work (January 1st, April 1st, July 1st, and October 1st). Eight clusters of physicians were included. Physicians were blinded regarding cluster randomization and the content of the other communication skills training. They gave written informed consent before undergoing three teaching modules of communication training (see section 2.2).”

Line 146-149, the period of assessment to be clearly stated.

Answer: We have added information on the period of assessment. Please find the updated wording below (page 7, lines: 165-175):

“Patients presenting to the ED were screened for the main complaint of chest or abdominal pain using the web-based electronic health record. The electronic health record was fed with information from the attending physician, showing the collected information, such as main complaint, almost in real time. If patients were eligible, trained study personnel explained the study procedure and informed consent was obtained right after patient history was obtained. Patients were blinded to the communication training which their physician had received. Information on demographics, mental and physical health (12-Item Short Form Health Survey; SF-12)(28), anxiety and depression (Hospital Anxiety and Depression Scale; HADS-D)(29) was obtained right after informed consent was given. Data were recorded using the web-based software secuTrial® by study personnel.”

Line 160, for Immediately discharged assessment, the period/time point to be clearly stated.

Answer: We have added information on the time point of discharge assessments. Please find the updated wording below (page 7, lines: 186-190):

“The primary outcome of the study was patients’ immediate recall of discharge information as a function of physicians’ communication training. This outcome was assessed right after the discharge communication was completed but before the patient left the hospital.”

Line 185 - 190, for the telephone conversation recording at 7 and 30 days and the method of recording to be clearly stated.

Answer: We have added information on the method of recording of the immediate recall but also of the follow-up telephone conversations. Please find the updated wording below (page7, lines: 186-191 & lines: 202-206):

“2.4.1 Primary outcome: Immediate discharge information recall. The primary outcome of the study was patients’ immediate recall of discharge information as a function of physicians’ communication training. This outcome was assessed right after the discharge communication was completed but before the patient left the hospital. The conversation between the researcher and the patient was recorded with an audio recorder.”

“Long-term recall of discharge information was assessed on day 7 and 30 after discharge via telephone interviews. The researcher was calling the patient on the phone at the time they agreed upon before the patient left the ED at day 0. The recording of the conversation started after obtaining the consent of the patient regarding the recording.”

Sample size calculation and statistical analysis

1 or 2 tailed test, sample size for each group and attrition rates consideration to be stated. The accepted level of significance to be stated.

Answer: Sample sizes are stated in the tables for each group and condition. Attrition rates are stated in Figure 2. We have added information on the test methods and accepted level of significance. Please find the updated wording below (pages 10-11, lines: 255-263). 

“The study was powered at 80% (two-sided test, �-level of .05) to detect a difference in recall performance for two patient groups discharged either by a physician trained in empathy or structured discharge communication. With an estimated effect size of .4 (as deduced from a previously conducted laboratory experiment) a total sample size of 200 was required. We used an alpha level of .05 for all statistical tests.

Analyses were conducted using R (version 3.6.1.) in RStudio using the packages tidyverse(32) and lme4(33) for calculating linear mixed effects models. For group comparisons, linear ANOVA analysis was used for continuous variables and Pearson’s two-tailed chi-square test of independence for categorical variables.”

Results

Line 256, what range refers to, to be clearly stated.

Answer: We have added information to clarify what the range referred to in line 256. Please find the updated wording below (page 12, lines: 296-299):

“80 physicians completed all three teaching modules during the study period, and 63 of them treated at least one patient within the context of this study (range: 1 to 19 patients per physician of which the discharge communication was recorded and included in the study).”

Table 1, at least one decimal point for percentages figures. Symbol <= , >= to be replaced with ≤, ≥ respectively. The decimal points for p values to be standardized. All the decimal points for the figures quoted in text to follow the exact figures in the table.

Answer: We have changed Table 1 accordingly: We added one decimal point for percentages, we replaced the symbols <= & >= with ≤ & ≥, and we standardized decimal points for p-values, as well as adapted and standardized decimal points in text, figures, and tables.

However, based on CONSORT guidelines, all statistical tests on group comparison at baseline to be avoided. As such all the description on the ‘statistical analysis’ on baseline to be avoided.

Answer: We agree that this is a recommendation in the CONSORT guidelines. Although, we randomized physicians to one of the two communication trainings, we wanted to ensure that clustering physicians to one condition did not lead to a systematic bias. Therefore, we thought it is important to report that the cluster randomization worked and did not bias the results. Nevertheless, if the reviewer feels strongly about this point, we are happy to remove the p-values and any references regarding group differences at baseline. 

Line 287 3.2.1., Line 302 3.2.2, Line 314 Table 2, the time point/period of assessment to be clearly stated.

Answer: We have amended the sections 3.2.1 (pages 15 & 16, lines: 396-397 & 410-411)); 3.2.2 (page 16, 420-421), and Table 2 (page 16, line: 438) to clarify that the time point of assessment that is mentioned refers to the time at discharge.

Line 332-337 & 342-355, results to be presented/tabulated in table form.

Answer: The manuscript already includes four tables. Given that there are only a few results presented in section 3.2.1, we would like to abstain from adding another table as an additional table might make it more difficult to digest the results for the reader. 

Line 334, 350, for substantial total explanatory power values, need to discuss more on the values and the reasons.

Answer: We have given more information on the values chosen and the reasons for these choices by providing the appropriate reference in the method section 2.5 (page 11, lines: 277-278).

Line 378-379, the p value for t test or correlation test to clearly separated.

Answer: We confirm that we are reporting the results of the correlation only. We improved the readability of the correlation reporting. Please find the exact wording below (page 24, lines: 503-504):

“In addition, adherence to recommendations within a month correlated significantly with immediate recall in the E group (r(81) = .29, p = .007) and the S group (r(58) = .32, p = .012).”

Line 383, for informativeness (p = .199), p value is 0.175. 0.199 is for patient affective index.

Answer: Thank you for this input. This is a mistake that we have corrected accordingly.

Line 385, p >0.124 to follow Table 4 p=0.124

Answer: We agree that the reporting of four different p-values like this can be confusing and therefore decided to now refer the reader to table 4, which is reporting the exact p-values of the PRA-D subscales (page 24, lines: 509-510).

Table 2, statistical test to be denoted in the table footnote.

Answer: We have denoted the statistical test in the table footnote, namely linear ANOVA (page 16: line: 439)

Table 3, Absolute, Relative, Range to be clearly denoted in the table footnote.

Answer: We have denoted all information necessary in the table footnote of Table 3.

Table 3 & 4, N- Miss to be clearly denoted in the table footnote or replaced with the word ‘missing’.

Answer: We have clearly denoted this information in the footnotes of Tables 3 & 4.

Effect size could be provided where applicable.

Answer: Effect sizes are reported for key results. If the reviewer is referring to specific additional results, we are happy to provide the effect sizes there as well.

A section to be provided to describe the missing data at overall and various time period i.e percentages etc.

Answer: Section “3.1.2 Patients” provides this information, please find the wording below (page 14, 381-386).

“In total, 1,915 patients were screened for eligibility (Figure 2). Of those, 1650 (80.0%) were not included because they did not meet inclusion criteria. A total of 265 patients were included in the study: 146 and 119 were treated by physicians of the E and S group, respectively. 196 (74.0%) patients completed the post-discharge assessment (111 in the E and 85 in the S group). Dropout rate from inclusion to post-discharge assessment was 26.0% (24.0% in the E and 28.6% in the S group).”

For the list of references, journal name and style to follow journal format.

Answer: Thank you for this input. We are now using “Vancouver” style for our references as recommended in the guidelines of PLOS ONE. 

Reviewer #3

This is an interesting study that tested the effectiveness of the structured discharge information presentation training relative to empathy training. The study was well conducted and the paper well written. The findings should be of interest to researchers and clinicians who are interested in intervention aimed to improve clinician-patient communication.

I have a few observations and minor suggestions for revisions.

1. There is no reason to think INFARcT training would be superior to empathy training on any outcomes other than recall and structure of information giving. The fact that information structure was superior to empathy was largely because the coding scheme was based on the INFARcT training itself. This is less about quality of information giving and more about fidelity of the intervention. I think this should be acknowledged in the discussion.

Answer: Thank you for this input. Indeed, the main point is about fidelity of the intervention. We have therefore amended the discussion accordingly. Please find the updated wording below (page 26, lines 542-546).

“Our manipulation check revealed that empathic communication skills were shown similarly in both training groups. Thus, our study evaluated the additional effect of focusing on structuring skills as a complement to communicating in an empathic way. This ensured that the comparison group was not obviously inferior to the explicit structure intervention.”

2. While there were differences in self reported adherence and recommending the physician, there is not theoretical reason provided for why this should be the case. While the study did not test a theoretical explanation, the authors should offer at least some speculation because I feel it inadequate to draw any conclusions on adherence (even if self-reported) and physician recommendation simply based on the assumption 'it must have been INFARcT training.'

Answer: This is an important point. There are some theoretical considerations that deal with the human need ‘to make sense of their life’. Some of them center around the term ‘coherent narrative’. Apparently, human beings want to identify a certain structure within a seemingly chaotic situation to have the impression of predictability. We did not include these more general concepts in the paper because we felt it would go beyond the research scope. However, there are some empirical findings by Robinson and colleagues (2012) (doi: 10.1016/j.pain.2011.02.010), Enzer and colleagues (2003) (doi: 10.1093/intqhc/mzg056) and a good review article that describe that patients, contrary to health care professionals, highly appreciate explicit structure, the elements of which have been described in detail by Gobat and colleagues (2015) (doi: 10.1016/j.pec.2015.03.024).

3. I would argue that the rationale for comparing INFARcT to empathy training is that empathy is considered the gold standard for effective communication is not convincing. Empathy is important, but quality of information exchange, perhaps shared decision-making (though less relevant in ED) are just as important. Thus, a better rationale needs to be provided (perhaps authors were interested in comparing an information intervention to a relationship-focused one). Or argue empathy as a control condition because it does not focus on information per se.

Answer: Many thanks for this suggestion. We tried to design a study with a control group that receives as much communication skills training as the information structuring group. Therefore, no formal training of residents in the control group was not an option for the study team. We agree that the role of empathy has been a matter of debate (Hoffstädt and colleagues (2020), doi: 10.1089/pmr.2020.0052; MacNaughton (2009), doi: 10.1016/S0140-6736(09)61055-2) and that labeling empathy training as the “gold standard” might lead to confusion. We therefore omitted this wording in the manuscript (i.e. in the introduction and discussion). We now provide our reasoning for choosing empathy, defined as the doctor’s ability to respond to emotions as a credible control group that does not focus on the provision of information per se (page 4, lines: 109-110):

“As a control group, some clusters of physicians were trained using empathy skills, ensuring a credible control group that does not focus on conveying information per se.”

4. I think the physician-patient relationship outcome is mislabeled. The PRA assesses patient perceptions of the quality of communication--physician informative, caring, and patient feels comfortable communicating. I suggest label it perceived quality of communication or perceived quality of relationship (i.e., needs 'quality' in the label).

Answer: You are correct. Therefore, we have therefore renamed the category to “perceived quality of relationship between patient and physician” (page 9, lines: 217-218).

5. In discussion, I think authors could say that the non information outcomes were acheived equally (for the most part) between the empathy and the information intervention, meaning patients value the quality of their information from providers as well as the quality of their relationships with providers.

Answer: This is an interesting aspect. We have added a sentence to the discussion about the “non information outcomes” that were achieved equally between the empathy and the information intervention. Please find the sentence below (page 27, lines: 570-573).

“Overall, our results show that the non-information outcomes were achieved equally well between the empathy and the information intervention, meaning patients value the quality of their information from healthcare providers as well as the quality of their relationships with providers.”

Thank you again for your valuable input. We believe that the manuscript has significantly improved by your thoughts and considerations. 

Kind regards,

Dr. Victoria Siegrist & Prof. Dr. Roland Bingisser, representing all authors of the manuscript

---

## [Decision Letter · Decision Letter 1]

17 Jun 2021

PONE-D-20-21645R1

Does Information Structuring Improve Recall of Discharge Information? A Cluster Randomized Clinical Trial

PLOS ONE

Dear Dr. Siegrist,

Thank you for submitting your manuscript to PLOS ONE. After careful consideration, we feel that it has merit but does not fully meet PLOS ONE’s publication criteria as it currently stands. Therefore, we invite you to submit a revised version of the manuscript that addresses the point raised by Reviewer #2.

We look forward to receiving your revised manuscript.

Kind regards,

Alessandra Solari, M.D.

Academic Editor

PLOS ONE

Journal Requirements:

Reviewers' comments:

Reviewer's Responses to Questions

**Comments to the Author**

1. If the authors have adequately addressed your comments raised in a previous round of review and you feel that this manuscript is now acceptable for publication, you may indicate that here to bypass the “Comments to the Author” section, enter your conflict of interest statement in the “Confidential to Editor” section, and submit your "Accept" recommendation.

Reviewer #1: All comments have been addressed

Reviewer #2: All comments have been addressed

Reviewer #3: (No Response)

2. Is the manuscript technically sound, and do the data support the conclusions?

Reviewer #1: Yes

Reviewer #2: (No Response)

Reviewer #3: (No Response)

3. Has the statistical analysis been performed appropriately and rigorously? 

Reviewer #1: Yes

Reviewer #2: (No Response)

Reviewer #3: (No Response)

4. Have the authors made all data underlying the findings in their manuscript fully available?

Reviewer #1: Yes

Reviewer #2: (No Response)

Reviewer #3: (No Response)

5. Is the manuscript presented in an intelligible fashion and written in standard English?

Reviewer #1: Yes

Reviewer #2: (No Response)

Reviewer #3: (No Response)

6. Review Comments to the Author

Reviewer #1: Thank you for addressing my concerns in this revision. I continue to feel that this is an important contribution. This time through I focused more on the fact that while the training helped with immediate recall, that effect faded over time. This suggests to me that you may want to more directly suggest some follow up structured action to ensure that there is appropriate follow up for ED patients.

Reviewer #2: The authors have put in great effort to address the comments.

Minor comments

Line 246, 346, the term linear ANOVA to be rephrased to reflect actual test name.

Reviewer #3: (No Response)

7. PLOS authors have the option to publish the peer review history of their article (what does this mean?). If published, this will include your full peer review and any attached files.

Reviewer #1: **Yes: **Eric Mark Eisenberg

Reviewer #2: No

Reviewer #3: No

---

## [Author Response · Author response to Decision Letter 1]

4 Jul 2021

Reviewer #1: 

Thank you for addressing my concerns in this revision. I continue to feel that this is an important contribution. This time through I focused more on the fact that while the training helped with immediate recall, that effect faded over time. This suggests to me that you may want to more directly suggest some follow up structured action to ensure that there is appropriate follow up for ED patients.

Answer: Thank you for your valuable input. We are convinced that the revised version has improved due to your feedback. We have now added a few sentences in the limitations section of the discussion suggesting some follow up structured action to ensure that there is appropriate follow up for ED patients (lines: 507-510): 

“Seventh, though training helped with immediate recall, the waning effect remains to be a concern. We therefore suggest a second structured information as a follow-up to boost recall and support patient adherence. Effects of such an additional structured intervention, however, should be studied in the future.”

Reviewer #2: 

The authors have put in great effort to address the comments.

Minor comments

Line 246, 346, the term linear ANOVA to be rephrased to reflect actual test name.

Answer: The term in lines 246 and 346 reflects the actual test name. Please note that Table 2 describes the differences in discharge communication by physicians. We chose linear ANOVA for group comparison as described in the methods section (lines: 245-246). Linear-mixed effects model analyses were used for the primary outcome in patients so that we could control for random factors such as cluster (lines: 249 – 257). Now all table footnotes include the name of the statistical test that was performed to ensure clarity.

With this rebuttal letter and the marked-up copy of our manuscript, we hope to answer all questions raised and would like to thank all reviewers and the editor once more for their valuable input. 

Kind regards,

Dr. Victoria Siegrist & Prof. Dr. Roland Bingisser, representing all authors of the manuscript

---

## [Editor Report · Decision Letter 2]

8 Sep 2021

Does Information Structuring Improve Recall of Discharge Information? A Cluster Randomized Clinical Trial

PONE-D-20-21645R2

Dear Dr. Siegrist :

We’re pleased to inform you that your manuscript has been judged scientifically suitable for publication and will be formally accepted for publication once it meets all outstanding technical requirements.

Kind regards,

Alessandra Solari, M.D.

Academic Editor

PLOS ONE

---

## [Editor Report · Acceptance letter]

7 Oct 2021

PONE-D-20-21645R2 

Does Information Structuring Improve Recall of Discharge Information? A Cluster Randomized Clinical Trial 

Dear Dr. Siegrist:

I'm pleased to inform you that your manuscript has been deemed suitable for publication in PLOS ONE. Congratulations! Your manuscript is now with our production department. 

Kind regards, 

on behalf of

Dr. Alessandra Solari 

Academic Editor

PLOS ONE